# Characteristics of Neurokinin-3 Receptor and Its Binding Sites by Mutational Analysis

**DOI:** 10.3390/biology10100968

**Published:** 2021-09-27

**Authors:** Ishwar Atre, Naama Mizrahi, Berta Levavi-Sivan

**Affiliations:** Department of Animal Sciences, The Robert H. Smith Faculty of Agriculture, Food, and Environment, Hebrew University of Jerusalem, Rehovot 76100, Israel; ishwar.atre@mail.huji.ac.il (I.A.); naama.mizrahi@mail.huji.ac.il (N.M.)

**Keywords:** CRE luciferase assay, molecular docking, neurokinin B, neurokinin F, neuropeptides, nonpeptide tachykinin receptor antagonist, spermatogenesis, Tac3Ra, tachykinin receptors, tilapias

## Abstract

**Simple Summary:**

This study presents in silico models of neurokinin B receptor tiTac3Ra (tilapia Tachykinin 3 receptor a) and its potential binding sites, as well as docking native tilapia Neurokinin F (tiNKF) and tilapia Neurokinin B (tiNKB) to theses orthosteric binding sites. For a better understanding of the binding confirmation and interaction of the structures, we compared the conformation between peptide docking and induced fit docking results. We have tried to analyze the affinity of binding and binding site interactions parallel to *in-vitro* results of receptor activity. NKB antagonists inhibit male tilapia gonadal development and gonadotropin release. We further verified the *in-vivo* effect of the antagonists on gonadal development in males. Studying the receptor activity of variants with alanine mutations at Phe251^6.44^ and Met289^7.43^ respectively, we found that, while both variants completely underperformed, there was no direct interaction with the ligand in the binding process indicating their role in post binding receptor activation rather than the binding process itself.

**Abstract:**

NKB (Neurokinin B) is already known to play a crucial role in fish reproduction, but little is known about the structure and function of NKB receptors. Based on an *in silico* model of the tilapia NKB receptor Tachykinin 3 receptor a (tiTac3Ra) found in the current study, we determined the key residues involved in binding to tilapia NKB and its functional homologue NKF (Neurokinin F). Despite studies in humans suggesting the crucial role of F251^6.44^ and M289^7.43^ in NKB binding, no direct peptide interaction was observed in tilapia homologs. *In-silico*, Ala mutations on residues F251^6.44^ and M289^7.43^ did not influence binding affinity, but significantly affected the stability of tiTac3Ra. Moreover, *in vitro* studies indicated them to be critical to tiNKB/tiNKF-induced receptor activity. The binding of NKB antagonists to tiTac3Ra both *in-vitro* and *in vivo* inhibits FSH (follicle stimulating hormone) and LH (luteinizing hormone) release and sperm production in mature tilapia males. Non-peptide NKB antagonist SB-222200 had a strong inhibitory effect on the Tac3Ra activation. SB-222200 also decreased LH plasma levels; two hours post intraperitoneal injection, changed sperm volume and the ratios of the different stages along the spermatogenesis in tilapia testes.

## 1. Introduction

Control of vertebrate reproduction is regulated mainly by the hypothalamic-pituitary-gonadal (HPG) axis. The hypothalamus contains several key neuropeptides-expressing neurons, such as gonadotropin-releasing hormone (GnRH) and kisspeptin neurons [1] that are crucial for the release of the gonadotropin’s (GTHs) follicle stimulating hormone (FSH) and luteinizing hormone (LH) [2]. Unlike mammals, teleosts do not possess a functional hypothalamus-pituitary portal blood system; rather, they have a dual mode of gonadotropin regulation by GnRH that combines both neuroglandular and neurovascular components [3]. Hence, the teleost’s proximal pars distalis (anterior pituitary) is innervated by neurons, synthesizing a number of neuropeptides and neurotransmitters involved in the regulation of the GTHs’ release (LH, and FSH release) [4]. In most teleost species, there are three GnRH types with multiple receptor types (GnRH-R). In the brain of the tilapia, GnRH1 (seabream GnRH) is the hypophysiotropic form since its neurons are present in the preoptic area and are primarily involved in the control of gonadotropin release. GnRH2 (chicken GnRH-II) and GnRH3 (salmon GnRH) neurons are present in the midbrain tegmentum and the terminal nerve, respectively, and they play neuromodulatory roles in reproductive and non-reproductive functions such as food intake and socio-reproductive behaviors [2,5,6,7]. Another unique feature of the teleost’s hypothalamus-pituitary is the special organization of the secretory cells in the pituitary [8], which makes fish a suitable model for studying neuroendocrine regulation. In recent years, several novel neuropeptides were identified as important regulators of GnRH synthesis and release in mammals and fish. Moreover, it was also shown that the loss of function mutations in the genes coding for these neuropeptides or their receptors, resulted in idiopathic hypogonadotropic hypogonadism (reviewed by [9,10,11]). It is reported that humans with inactivating mutations in GPR54 (G-Protein coupled receptor-54), the cognate receptor for kisspeptin, exhibit normosmic hypogonadotropic hypogonadism [12,13]. This identification of a novel neurocircuit upstream to the GnRH signaling has revolutionized the field of reproductive endocrinology [14]. However, while kisspeptin is a key regulator of GnRH release in mammals [15], its role in teleosts remains controversial, especially since knockout of the two kisspeptin and two KissRs genes in zebrafish (*Danio rerio*) and medaka (*Oryzias latipes*) had no obvious effect on their reproductive capability [16,17]. Similar to kisspeptin, the loss-of-function mutation in neurokinin B (NKB; also known as tachykinin 3 (Tac3)) or of its receptor (NKBR, Tac3R, NK3R) also exhibit hypogonadotropic hypogonadism in humans [9]. NKB belongs to the tachykinin family, a large family of neuropeptides with various biological functions and widely conserved among invertebrates and mammals [18]. The tachykinin family includes neuropeptides such as NKB, neurokinin A (NKA), substance P (SP) etc. All neuropeptides in this family exhibit a common C-terminus tail sequence, Phe-X-Gly-Lys-Met-NH_2_, where X is a hydrophobic amino acid [19]. Unlike mammals, the teleost’s tac3 gene encodes two peptides, NKB and NKB-related peptide, also termed neurokinin F (NKF) and two tac3 receptor genes (Tac3Ra and Tac3Rb) [20]. NKB is vividly expressed in fish species and was shown recently to have a stimulatory reproductive effect in zebrafish, [20], goldfish (*Carassius auratus*) [21], tilapia (*Oreochromis niloticus*) [22], orange-spotted grouper (*Epinephelus coioides*) [23], eel (*Anguilla Anguilla*) [24], and the striped bass (*Morone saxatilis*) [25].

In mammals, the arcuate nucleus contains a unique population of neurons that co-express kisspeptin, NKB and the opioid dynorphin and are now collectively referred to as KNDy neurons [26,27,28]. Mammalian KNDy neurons coordinately activate the GnRH pulse generator [29]. In fish, NKB was shown to stimulate the release and synthesis of LH and FSH both *in-vivo* and *in-vitro* from several species such as zebrafish [20], tilapia [22,30], and striped bass [25]. However, in several teleost fish, such as medaka (*Oryzias latipes)* [17,31], goldfish (*Carassius auratus*) [17], European sea bass (*Dicentrarchus labrax*) [32] and African cichlid (*Astatotilapia burtoni*) [33], studies have shown that GnRH neurons do not possess kisspeptin receptors. Interestingly, GnRH neurons do possess NKB-R in tilapia [30], but are absent in sea bass GnRH neurons [25]. Moreover, FSH- and even more, LH cells possess tac3r, indicating a direct effect at the pituitary levels [22,30,34]. NKB analogs induced mRNA synthesis of all three GnRH types in the brain, and NKB receptors were co-expressed in all three GnRH neuronal types in different regions of the brain, suggesting that besides the known direct effect of NKB on NKB receptors at the pituitary level [22], there is an indirect effect through secretion of GnRH that ends with increased secretion of gonadotropins. Taken together it seems that the NKB/NKBR system in fish is an important reproductive regulator in a similar way to the kisspeptin system in mammals.

The Tac3Rs are peptides receptors belonging to the class A (Rhodopsin) family of GPCR consisting of an extracellular domain (EXD), seven transmembrane domains (TM) and intracellular cytosolic domains (ICD) [35]. The synthetic zebrafish and tilapia NKBs activated Tac3 receptors via both PKC/Ca^2+^ and PKA/cAMP signal-transduction pathways *in-vitro* [20,22,30]. As to our knowledge thus far, there is no available crystal structure for Tac3Rs.

Tilapia is one of the most important fish in aquaculture and an emerging model system for laboratory studies in many fields [36]. It is already known that the NKB system in tilapia plays an important reproductive role [22] but little is known about the structure and functionality of the NKB receptors. In the current work, we have investigated the structure of tilapia Tac3Ra and studied the functionality of this receptor using specific non-peptides NKB antagonists and mutagenesis *in-vitro* and *in-vivo*. We have studied the binding site and binding interaction of tiNKB and tiNKF to tiTac3Ra. We have further focused our study to understand the role of Phe251^6.44^ and Met289^7.43^ in receptor activation. We have further performed Ala mutation on these residues and observed *in-vitro* effects on receptor activity in response to native peptides and their antagonists. We have also confirmed the role of the selected antagonists on the spermatogenesis in mature adult male tilapia.

## 2. Materials and Methods

### 2.1. Fish Maintenance and Growth

Nile tilapia (*Oreochromis niloticus*) were kept and bred in the fish facility at the Hebrew University in 500-L tanks at 26 °C, with 14 h light and 10 h dark photoperiods. The fish were fed daily ad libitum with commercial fish pellets (Raanan fish feed), which has 6% fat and 50% protein. All experiments were conducted on sexually mature fish. All experimental procedures were in compliance with the Animal Care and Use guidelines of the Hebrew University, and were approved by the local Administrative Panel on Laboratory Animal Care.

### 2.2. NKB & NKF Peptide Synthesis

Tilapia NKB (pyroglutaminated [p]-EMDDIFIGLM-NH2) and tilapia NKF (YNDLDYDSFVGLM-NH2) [22] were synthesized by GL Biochem (China). Peptides were synthesized by the automated solid-phase method by applying Fmoc active-ester chemistry, purified by HPLC to >95% purity [37] and the carboxy terminus of each peptide was amidated. The peptides were dissolved to the desired concentration in fish saline (0.9% NaCl in double distilled water) for *in-vivo* experiments, and in culture media for *in-vitro* experiments as per previous studies [22,30].

### 2.3. Structure Prediction, Homology Modelling and Molecular Docking

An in silico model for tilapia NKB receptors, the potential binding pocket, and B-factor prediction for model stability, were prepared on the I-TASSER server [38,39], using available models for substance P receptors (PDB:2KS9, PDB:6J21 and PDB:6HLL) as the template for sequences (NCBI Ref seq: NP_001288307). We employed the threading assembly refinement (I-TASSER) [40,41,42] method to generate structure predictions for tiTac3Ra with available models of similar proteins as a template. Estimation of the distance errors was conducted using support vector regression utilizing convergence of threading alignment, divergence of I-TASSER simulation decoys, and sequence-based secondary structure and solvent accessibility predictions. The domain prediction was implemented using CBS prediction tools, and phosphorylation sites for serine, threonine, and tyrosine were predicted using the NetPhos 3.1 server [43]. The membrane protein topology models and potential N glycosylation sites were generated using the Protter online tool [44]. The trans-membrane domain prediction was performed using the TMHMM online tool [45].

The tiNKB and tiNKF peptide models were generated with available huNKB (PDB:1P9F) as the template developed by [30], while antagonist models for Talnetant (Pubchem CID: 5311424), Osanetant (Pubchem CID: 219077), SB-222200 (Pubchem CID: 6604009) were taken from the Pubchem database. The NKB antagonist ligands were prepared using the Schrödinger (Schrödinger Release 2021-2: LigPrep, Schrödinger, LLC, New York, NY, USA, 2021), and returned 2, 6, and 6 stereoisomers for SB-222200, Osanetant and Talnetant, respectively.

We further used the Consurf server [46] to estimate the conservation of residues among the closest homologs. Further processing of the structures, binding site prediction, docking and mutation analysis were done with Schrodinger (BioLuminate, Schrödinger, LLC, New York, NY, USA, 2020). The receptor grid was generated in accordance with the predicted binding sites. We docked native tiNKB, tiNKF and antagonists, SB-222200, Osanetant and Talnetant were docked at Binding Pocket AB and Binding Pocket C. No docking poses were returned on docking to Binding Pocket C. A mutation analysis was conducted for the selected docked poses, with Alanine replacing the receptor amino acids one at a time. (Spreadsheet S1 & Spreadsheet S2).

### 2.4. Construction of F251A and M289A tiTac3Ra Expression Vector

The expression vector for tilapia tiTac3Ra (GenBank accession no. KF471674) [22] was modified with the Stratagene QuikChange mutagenesis kit (Stratagene, La Jolla, CA, USA), according to the manufacturer’s instructions using two complementary primers (Table 1). The procedure included 18 PCR cycles as specified by the manufacturer’s manual, and the use of Pfu polymerase for the reaction. The mutated construct was then digested with the restriction enzyme *DpnI*, which is specific to methylated and hemi-methylated DNA, to digest the template and to select the mutation-containing synthesized DNA. The vector was then transfected into XL1 competent cells (*E.coli*). Two colonies were sequenced and confirmed to contain the mutation without any undesired misincorporation of nucleotides.

### 2.5. Construction of tiTac3Ra Fused to GFP

The entire open reading frame of the wild-type and the mutant forms of tiTac3Ra were cloned into pEGFP-N1 (Clontech, Palo Alto, CA, USA). In order to fuse the receptor to the reading frame of the green fluorescent protein (GFP), PCR was performed using the Reverse primer AAAAAACTGCAGAGGGAATTCTTCAGGCTGAGC, in which the receptor’s stop codon was eliminated and replaced with the amino acid glycine, and a *PstI* site was inserted (underlined). The receptor was sub-cloned into pGEM-T Easy (Promega, Madison, WI, USA). The fragment was then Fast Digest by *XhoI* and *PstI* and cloned into the same site of the expression vector pEGFP-N1. Ligation products were transformed into XL1 competent cells. Clones were verified by sequence analysis (Sequencing Unit, The Weizmann Institute of Science, Rehovot, Israel).

### 2.6. Receptor Transactivation Assay

In order to study the signaling pathways of the mutant tiTac3Ra, the entire coding regions of tiTac3Ra (GenBank accession no. KF471674), and both of the mutant receptors were inserted into pcDNA3.1 (Invitrogen, Massachusetts, United States). We used a sensitive luciferase (LUC) reporter gene assay. We previously demonstrated that cAMP response element (CRE-LUC; Invitrogen, Massachusetts, United States) reporter systems are a useful tool for discriminating cAMP/PKA signaling pathways [47], and that PKA is the dominant pathway for tiTac3Ra [20,22]. Transient transfection, cell procedures and stimulation protocols followed the procedures previously described [48]. Briefly, 3 μg of each construct, together with 3 μg of a luciferase reporter plasmid were transiently transfected into COS-7 cells (American Type Culture Collection, Manassas, VA, USA). Forty-eight hours after transfection, cells transfected with Tac3Ra plasmid fused with GFP were applied with an Hoechst dye (Invitrogen, Massachusetts, United States; diluted 1:1000) in order to mark nuclei in the culture for transfection analysis.

Cells transfected with the tiTac3Ra construct and luciferase reporter plasmid were treated with tilapia NKB (tiNKB) and tiNKF at increasing doses (0.2–1000 nM). Luciferase activity (Luc) was measured, and compared to basal treatment, and the maximal effect (Luc fold activation) was calculated for each treatment. Only results with a threshold of >2.0 were calculated for their IC50 values, in order to avoid false values.

Non-peptide molecules, SB-222200 (Sigma, Merck, Darmstadt, Germany) and Osanetant (Axon Medchem, Reston, VA, USA), known to antagonize mammalian NK3R [49] were studied for their ability to antagonize tiTac3Ra, which was activated with a fixed dose of the native ligands (10 nM tiNKB or tiNKF), combined with increasing doses of the antagonists (0.001–1000 μM). The antagonists were added 10 min prior to the addition of the peptides. IC_50_ values were calculated from concentration response curves by means of computerized nonlinear curve-fitting with Prism version 9 software (GraphPad, San Diego, CA, USA). All experiments were repeated at least 3 times using cells from independent transfections, each performed in triplicate.

### 2.7. ELISAs for the Measurement of Tilapia FSH LH and GH

The levels of FSH, LH and GH in the plasma were measured by specific competitive enzyme-linked immunosorbent assay (ELISA), developed for tilapia [50], based on recombinant gonadotropins or GH that were created by using the yeast *Pichia pastoris*. For the ELISAs, we used primary antibody antisera against GH [51], FSHβ [52], or against rtLHβ [53], and rtFSHβα [52], rtLHβ [53] or tiGH for the standard curves. Sensitivity for the plasma measurements were 15.84 pg/mL for LH, 0.24 pg/mL for FSH and 30.0 pg/mL for GH. The inter-assay coefficient of variation (CV) was 14.8, 12.5, and 13.0%, while intra-assay CV was 7.2, 8.0, and 8.0% for LH, FSH and GH, respectively.

### 2.8. In Vivo Effect of Non-Peptide NKB Antagonists

Adult male tilapia (BW 89.29 ± 32.93 g; gonado-somatic index (GSI) 0.24% ± 0.40%) were injected IP (intraperitoneal) with NKB antagonists, SB-222200 or Osanetant (100 µg/kg BW), and after 0.5 h, a second injection of sGnRHa (10 µg/kg BW) was added. Control groups were injected with sGnRHa ([DAla^6^, Pro^9^-Net]-mammalian GnRH; Bachem, Budendorf, Switzerland) 10 µg/kg BW) as a positive control. The fish were bled from the caudal blood vessels into heparinized syringes 0, 0.5, 2, 6 and 24 h after injection. This time course is according to standard protocols used previously [52,54] to test the effect of GnRH on circulating levels of LH, FSH, and GH in tilapia. Blood samples were collected from the caudal vasculature and centrifuged (3200 rpm for 30 min at 4 °C) (Centrifuge: Fresco 17, Rotor: 75003424, Thermo Scientific™, Langenselbold Germany) to obtain plasma samples, which were stored at −20 °C until assayed. Three independent experiments were carried out for the *in-vivo* studies.

### 2.9. In Vivo Effect of NKB Antagonist on Sperm Production in Tilapia

Adult male tilapia (BW 87.29 ± 26.1 g; gonado-somatic index (GSI 0.21% ± 0.36%) were IP injected with SB-222200 (500 µg/kg BW in 25% DMSO) and saline with 25% DMSO as a control every 48 h during a 14-day period. Blood was sampled every 48 h. Milt volume was collected from 6 fish from each group 7 and 14 days after the first injection, generally according to [55].

### 2.10. Testicular Histology and Quantification of Cell Composition

Testes were sampled for histology at 14 days, and testicular cell composition was analyzed. Testes were fixed for 24 h, after which a portion was removed (~5 mm in length) from the center of the larger right testicular lobe for histological processing. Samples were rinsed in distilled water, dehydrated in an alcohol series, cleared in xylene, embedded in paraffin, serially sectioned at 4 μm in the transverse plane, and stained with hematoxylin and eosin using standard histological techniques. Photomicrographs were then taken (4× objective) of 3 randomly chosen sections from each individual for analysis of testicular cell composition. Randomly generated points (*n* = 50) were overlaid on each photograph (CPCe software) [56], and the cell type underneath each crosshair point was identified. A minimum of 50 points was necessary to adequately determine the percentage of cell types. Cell types were identified based on previous descriptions in this and other related teleost species reviewed in [57], and the following cells were quantified: spermatogonia, spermatids, and spermatozoa (mature sperm). Each cell type was then expressed as a percentage of the total, and averaged across the 3 sections to obtain a mean for each individual.

### 2.11. 11-Ketotestosterone (11-KT) Analysis

11-KT levels were determined by ELISA, according to [52], using acetylcholine-esterase as a label. The anti- 11-KT was donated by Dr. D.E. Kime (Sheffield, UK). All samples were analyzed in duplicate, and for each ELISA plate, a separate standard curve was run. The lower limits of detection were 0.93 for 11-KT. The intra- and inter-assay coefficients of variance were less than 7%. Steroid levels in the plasma, determined by ELISA, were validated by verifying that serial dilutions were parallel to the standard curve.

## 3. Results

### 3.1. Homology Modelling

For the current study, the optimum predicted model was selected based on high confidence (C-Score), high structural stability, and similarity between templates and the query sequence [41,42]. To our understanding, this is the most stable structure for the query sequence (Figure 1A,B). The receptor structure was observed to be a peptide receptor of the Class A (Rhodopsin) family GPCR [58]. The selected tiTac3Ra model had a C-score of 0.17; a TM score of 0.69 ± 0.12 and its RMSD was 7.3 ± 4.2 Å. 3-D models for native peptides tiNKB (Figure 1C) and tiNKF (Figure 1D) were generated and further refined using Schrodinger (BioLuminate, Schrödinger, LLC, New York, NY, USA, 2020). The comparison of the superposition of the generated models to huNK1R (PDB:6J21; PDB:6HLL) [59,60] and huNTSR1 (neurotensin receptor) (PDB:4GRV) [61] suggests the model to be in inactive conformation.

The peptide structures of NKB and NKF were seen to be unidirectional α-helix loop-rich peptides, pointing toward the amide NH groups, with hydrogen bonds between every 4th amino acid (5.4 Å) as reported in previous studies [30]. The structure for NKB antagonists SB-222200 (Figure 1E), Osanetant (Figure 1F) and Talnetant (Figure 1G) were obtained from the PubMed database and further refined and returned 2, 6, and 6 stereoisomers for SB-222200, Osanetant and Talnetant, respectively.

### 3.2. Binding Site Detection

Major binding pockets were predicted in the extracellular region and transmembrane domain, typical of the peptide receptor family of Class A, GPCR (Figure 2A,B). Binding sites were selected for docking NKB and NKF peptides and NKB antagonists on the basis of their apparent binding locations on receptors on the human tachykinin homologs [59,62,63,64] and site scores. For this study, the larger pocket was labelled Pocket A, and was localized at the extracellular regions and the hydrophobic cleft formed of Pro105^3.32^, Val106^3.33^, Val109^3.36^, Val94^3.21^, Val198^5.46^, Trp255^6.48^, Tyr258^6.51^, Phe262^6.55^, Tyr281^7.35^, Trp286^7.40^, Ala288^7.42^, Met289^7.43^ in the upper transmembrane domain (Figure 2A; Table 2). The smaller pocket was labeled B (Figure 2B; Table 2), and was buried deep within the transmembrane domain. Another orthosteric sub-binding pocket was predicted for the binding of the antagonist in this cleft (Table 2) sharing a majority of residues with the aforementioned Binding Pocket A, and positioned similar to the antagonist binding pocket reported in huNK1R [60].

### 3.3. Peptide Docking and Induced Fit Docking

The reported docking for all the peptides and antagonists was carried out on all three Binding Pockets. No poses were returned for docking on Pocket B, despite showing a high site score, suggesting it can be a potential binding site for smaller compounds, but it is too small for those used for the native ligands used in this study. We conducted both peptide docking and IFD (Induced fit docking) of the tiNKB and tiNKF peptides on a generated model in order to study their binding affinity to tiTac3Ra (Figure 3), and their molecular interactions. The peptide docking module returned a total of 100 and 98 predicted binding poses for tiNKB and tiNKF peptides, respectively. Further screening was conducted on the basis of conformational probability, Glide emodel (kcal/mol), Glide score (gscore) (kcal/mol), and interaction fingerprinting analysis. The peptide models were selected after considering their Glide emodel scores (tiNKF: −128.848 kcal/mol and tiNKB: −135.172 kcal/mol), (tiNKF: −9.901 kcal/mol and tiNKB: −12.211 kcal/mol) (Appendix A and Spreadsheets S3 and S4) and receptor ligand interaction. The selected IFD docking model were selected similarly Glide emodel scores (tiNKF: −337.278 kcal/mol and tiNKB: −182.706 kcal/mol) and gscore (tiNKF: −18.254 kcal/mol and tiNKB: −13.106 kcal/mol) and receptor ligand interactions (Table 3 and Table 4). The 6th TMD helix in the IFD of NKF showed a larger outward deviation in comparison to NKB IFD (RMSD: 2.4156).

**Table 3 biology-10-00968-t003:** IFD: Major Specific Binding interactions between tiTac3Ra and tiNKF (Refer Figure 2).

Residue on NKF	Specific Interactions with Residue on tiTac3Ra	Position on tiTac3Ra
B:1:Asp	1× hb, 1× salt bridge to A:273:Lys	EXL3
B:3:Asp	1× hb, 2× clash to A:17:Gln	EXD
	1× hb, 1× salt bridge to A:273:Lys	EXL3
B:4:Tyr	1× hb to A:14:His	EXD
	1× hb to A:17:Gln	EXD
	2× clash to A:90:Glu	EXL1
B:5:Asp	2× hb, 1× clash to A:173:Arg	EXL2
	1× salt bridge, 2× clash to A:273:Lys	EXL3
B:6:Ser	1× hb to A:85:Tyr	TM2
B:7:Phe	1× pi stack to A:177:Tyr	EXL2
B:9:Gly	1× hb to A:281:Tyr	TM7
B:11:Met	1× clash to A:258:Tyr	TM6

hb = hydrogen bond.; EXD = Extra cellular domain; EXL = Extra cellular Loop; TM = Transmembrane helix.

**Table 4 biology-10-00968-t004:** IFD: Major Specific Binding interactions between tiTac3Ra and tiNKB. (Refer Figure 4).

Residue on NKF	Specific Interactions with Residue on tiTac3Ra	Position on tiTac3Ra
B:3:Asp	1× hb, 1× clash to A:17:Gln	EXD
	1× clash to A:170:Ile	EXL2
	1× salt bridge, 1× clash to A:273:Lys	EXL3
B:4:Asp	1× hb to A:173:Arg	EXL2
	1× hb to A:273:Lys	EXL3
B:5:Ile	2× clash to A:272:Ser	EXL3
B:6:Phe	1× hb to A:173:Arg	EXL2
	1× hb to A:272:Ser	EXL3
B:7:Ile	1× clash to A:18:Phe	EXD
	1× clash to A:173:Arg	EXL2
B:8:Gly	1× hb to A:177:Tyr	EXL2
B:9:Leu	1× clash to A:85:Tyr	TM2
	1× hb to A:281:Tyr	TM7
B:10:Met	1× hb to A:177:Tyr	EXL2
	2× clash to A:261:Tyr	TM6
	4× clash to A:262:Phe	TM6
	1× clash to A:265:Thr	TM6

hb = hydrogen bond.; EXD = Extra cellular domain; EXL = Extra cellular Loop; TM = Transmembrane helix.

### 3.4. Residue Mutation Analysis

A residue mutation analysis was conducted to predict the effect of Alanine mutations on the stability and binding affinity of both the native peptides and tiTac3Ra (Appendix A, Spreadsheets S1 and S2). The binding affinity between tiNKB/tiNKF and tiTac3Ra was noted to increase by Ala mutations at Asp4Ala and Gly8Ala of the tiNKB, Asp5Ala of tiNKF. Ala mutations on all the other amino acids showed decreases in Δ Affinity. Ala mutation at M74^2.53^, Asn78^2.57^, Asn82^2.61^, Ile175^E2.49^, Phe251^6.44^, Tyr258^6.51^ Met285^7.39^, and Met289^7.43^, Arg173, Tyr281, reduced stability of the receptor-peptide complex significantly with the exception of Tyr257^6.51^ which showed a significant increase in the receptor stability in the case of binding with tiNKF.

### 3.5. In Vitro Luciferase Assay for Peptide Induced Activity

In order to evaluate and compare the transfection efficiency of the wild-type native tilapia Tac3Ra (WT) and both mutants (Phe251^6.44^Ala and Met289^7.43^Ala), we used a plasmid that encodes GFP as an internal control for the direct measurement of transfection efficiency in transient transfection assays. Transfections with the tilapia Tac3Ra-WT, -Phe251^6.44^Ala and -Met289^7.43^Ala expression vectors detected comparable transfection efficiencies in COS-7 cells (16.3 ± 0.03%, 22.9 ± 0.02% and 19.4 ± 0.07% respectively; Appendix A). Luciferase activity was measured, compared to basal treatment, and the maximal effect (Luc fold activation) was calculated for each treatment (Table 5). Native tilapia Tac3Ra (WT) and both mutants (Phe251^6.44^Ala and Met289^7.43^Ala) were activated by tiNKB (Figure 5A) and tiNKF (Figure 5B) dose-dependently (0.2–1000 nM) in CRE-Luc receptor activation analysis. Receptor transactivation for the binding of native tiNKB and tiNKF to tiTac3Ra and its mutant homologs (Phe251^6.44^Ala, Met289^7.43^Ala) showed significant differences in the comparative CRE-luciferase activity. A substantial decrease in receptor activity was observed in the case of both mutants.

### 3.6. Antagonist Binding and Activity

#### 3.6.1. In Silico Binding of Antagonist to tiTac3Ra

To further develop our insight towards the role of Phe251^6.44^ and Met289^7.43^ and see the effect of the Ala mutation to antagonist activity we used known NKB antagonists SB-222200, Osanetant and telanetant. Similar to the native peptides, the three antagonists were docked to Site A and B individually. Docking at Binding Pocket A returned 37 poses; no poses were returned on Binding Pocket B. The poses were screened in a similar manner to that used for the peptides. The models were selected after considering their Glide score (Osanetant: −9.198 kcal/mol; Talnetant: −9.633 kcal/mol; SB-222200: −6.792 kcal/mol) and Glide emodel (Osanetant: −71.776 kcal/mol; Talnetant: −47.079 kcal/mol; SB-222200: −48.197 kcal/mol) (Figure 6). Osanetant showed π-π stacking interactions to Phe262^6.55^ and His191^5.39^, and halogen bonds to Tyr85^2.64^ on tiTac3Ra. Talnetant showed π-π stacking interactions to Phe262^6.55^ and Tyr258^6.51^, whereas SB-222200 showed π-π stacking interactions to Phe262 and H-bond interactions with Tyr258^6.51^. No interaction was seen with Phe251^6.44^ and Met289^7.43^. In the human homolog NK3R, only the Me-talnetant variant of the Talnetant was reported to have binding interactions with tiTac3Ra Met289^7.43^ and homolog huM346^7.43^, whereas Osanetant and Talnetant showed no direct interactions [65,66].

#### 3.6.2. In Vitro Luciferase Assay of Antagonist Activity

Taking the strong similarity between the structures of Talnetant and its variant SB-222200 into account, only SB-222200 and Osanetant were used for *in-vitro* and *in-vivo* studies. NKB antagonists, SB-222200 (Figure 7A) and Osanetant (Figure 7B) were added in increasing concentrations (0.001–1000 μM) to transfected COS-7 that expressed tiTac3Ra. Ten minutes after the induction of the antagonists, the ligand, tiNKB or tiNKF (10 nM) was added to the media. Both antagonists had a strong inhibitory effect on the tiTac3Ra. SB-222200 was more efficient with tiNKB and tiNKF (IC_50_ = 84.11 and 37.05 μM, respectively) than was Osanetant (IC_50_ = 211.7 and 121.7 μM, respectively). Additionally, the luciferase activity, using SB-222200, was reduced to 50% of its maximal response, whereas it was reduced to only 75% with Osanetant. SB-222200 also had a higher maximum response rate than Osanetant for tilapia (Figure 7C,D).

As with the docking of peptides to receptors, a similar loss of CRE-LUC activity was observed. The intial level of activity, as seen in the wild-type, was absent in the Phe251^6.44^Ala and Met289^7.43^Ala mutants.

#### 3.6.3. In Vivo Effect of Antagonist Treatment

To determine the effect of the NKB antagonists SB-222200 and Osanetant on the release of gonadotropins, we conducted an in vivo experiment using male tilapia. NKB antagonists were given as initial IP (Intraperitoneal) injections at time 0, and half an hour later sGnRHa was injected into the same groups. Two hours after the injection, LH plasma levels were significantly lower after SB-222200 treatment than in response to sGnRHa (Figure 8A). A combination of SB-222200 and sGnRHa also had an inhibitory effect on LH release, without any significant change when compared to the sGnRHa. On the other hand, Osanetant had no inhibitory effects on LH plasma levels, when applied with GnRHa (Figure 8A). All treatments increased FSH plasma levels (Figure 8B) compared to the control group at six hours post injection. The combination of both NKB antagonists with sGnRHa increased FSH levels at 2 h post injection.

An additional in vivo experiment was conducted in order to investigate the long-term effect of the NKB antagonist, SB-222200, on gonadal development and sperm production. Gonadal development and sperm production are energy-intensive processes. The precise interaction of regulators for energy balance and reproduction enables coordinated regulation of these two processes [67]. Male tilapia were IP injected with SB-222200 (500 µg/kg BW) every 48 h within a 14-day period. Plasma GH and gonadotropins did not change significantly during the 14 days of the experiment, compared to those of control group (Appendix A). The levels of 11-KT rise with and in turn aids in spermetogenesis; however, agonist adminisration significantly delayed the rise of 11-KT levels in plasma (Figure 9A). Milt volume was collected from six fish from each group on the 7th and 14th day after the first injection. The fish treated with SB-222200 retained a low sperm production after 7 and 14 days compared to that of the control group. The latter showed a higher milt volume after 14 days compared to their milt production 7 days previously (Figure. 9B). The testes were sampled for histology at 14 days, and the testicular cell composition was analyzed (Figure 9C–F). The spermatogonia cell type did not differ between the control group and those treated with the NKB antagonist (32.87% and 35% respectively; Figure 9E). However, the more advanced cell types, namely spermatid and spermatozoa, were significantly reduced in fish treated with SB-222200 compared to those in the control group (10%, 6.63% and 23.2%, 23.78%, respectively). The NKB antagonist changed the proportion of the different stages along the spermatogenesis in the tilapia testes and significantly reduced the number of spermatozoa, indicating that the injected fish were less fertile.

## 4. Discussion

As there is no available crystal structure for neurokinin B receptors, the underlaying mechanism of NKB is neither clear nor well understood. Using in silico, *in-vitro*, and *in-vivo* methods to predict model structure and binding sites, we tried to understand the binding mechanism with its native ligands and available antagonists in the tilapia. Although tertiary structural information is crucial for function annotation, as to our knowledge, no Crystal NMR (neuclear magnetic resonance) or cryo-EM (Elecro magnetic) structure is available for tiTac3Ra or its homlogs in other vertebrate species. The presented receptor model is generated using available huNK1R receptor [59,60,68] templates. The tiTac3Ra as observed belongs to the class A GPCR (Rhodopsin) family [58,63,69] and exhibits seven transmembrane domains (TM1-7) sequentially connected by intracellular (ICL1-3) and extracellular loops (EXL1-3) and an extracellular (EXD) and a cytosolic domain containing amphiphatic helix 8 (ICD) accordingly. Extracellular tips of TM1, 4, 5 and 7 TM helices diverged towards the central GPCR transmembrane axis forming the binding cavity. The presented model expressed a high confidence score (C-score), and to our understanding, the best model from the predicted cluster and was further used for docking of the ligands.

We then selected two potential binding pockets from the detected binding sites named Binding Pocket A and B, respectively. Binding Pocket B was localized to be deep within the transmembrane domain but did not return any poses on docking despite having a high site score, but may prove to be a potential target for smaller compounds to manipulate tiTac3Ra activity. Structural alignment to huNK1R(PDB:6J21 & PDB:6HLL) [59,60] showed overlap of Binding Pocket A of tiTac3Ra to the antagonist binding pockets of NK1R in the hydrophobic cleft (Figure 2). Despite being closely related, unlike Substance P binding to huNK1R [68], tiNKB and tiNKF do not seem to bind superficially on the extracellurar region. Our models suggest Phe-X-Gly-Lys-Met-NH_2_ tail of the neurokinins binds deep in the hydrophobic cleft of the tiTac3Ra Binding Pocket A. In tiTac3Ra, hydrogen-bond interactions were observed between Asp3:Gln17, Tyr4:his14, Asp5:Arg173, Ser6:Tyr85^2.64^, Gly9:Tyr281, Met11:Tyr258^6.51^ and π-π stacking on Phe7:Tyr177, on ECL2; of tiNKF: tiTac3Ra complex (Table 3) and Asp3:His17; Asp4:Arg173; Ile5:Ser272; Phe6:Arg173, Ser272; Asp8:Tyr177; Leu9:Tyr281; Met10:Tyr177 of tiNKB:tiTac3Ra complex (Table 4). Sulfide bridges were observed with Arg273 in both tiNKF: tiTac3Ra and tiNKB:tiTac3Ra complex. Previous studies on the human homolog NK3R reported that Phenylalanine at the 6th aa position on human NKB (huNKB Phe6) binds to hydrophobic pockets Ser130^2.49^, Pro165^3.32^, Ile166^3.33^, Ala168^3.35^, Val169^3.36^, Phe170^3.37^ and Trp132^2.51^. Furthermore, Phe6 was noted to be optimally oriented to interact with the phenyl ring system of Trp312^6.48^ and Phe170^3.37^ [63]. Leu9 of huNKB bound to the pockets formed by Cys311^6.47^, Trp312^6.48^, Pro314^6.50^, Ala345^7.42^, Met346^7.43^ and Ser347^7.44^, whilst Met10 of huNKB bound to Phe123^2.42^, Val169^3.36^, Ser172^3.39^, Met176^3.34^, Val304^3.40^, Phe308^6.44^, Cys3116.47, Ser347^7.44^, Ser348^7.45^ and Met350 [63]. Our Binding Pocket A in tiTac3Ra overlaps with a cavity formed by residues Pro105^3.32^, Val106^3.33^, Val109^3.36^, Val94^3.21^, Val198^5.46^, Trp255^6.48^, Tyr258^6.51^, Phe262^6.55^, Tyr281^7.35^, Trp286^7.40^, Ala288^7.42^, and Met289^7.43^, and displayed a similar docking confirmation. The terminal C-terminus tail of the peptide, with the amino acid sequence Phe-X-Gly-Leu-Met-NH2 docked to the hydrophobic cleft in the transmembrane pocket. In the IFD confirmation in the case of tiNKF, the extracellular tips of the TMD 6 on ti Tac3Ra making a broader binding pocket to accommodate tiNKF hence showing a greater affinity for it. Despite NKF showing higher binding affinity in IFD docking NKB show better potency suggested higher binding potential of NKB to the native receptor and is strongly supported by previously reported *in-vitro* data in tilapia [22] and zebrafish [20]. Our *in-vitro* results suggest that the EC50 for tiNKF (65.22 ± 18.43) is several magnitude higher than that of tiNKB (2.48 ± 0.05) suggesting tiNKB as a more efficient ligand to tiTac3Ra. The better response to tiNKB needs better understanding towards post docking molecular dynamics of these complexes.

Being a larger structure, the magnitude of factors that can affect the binding affinity and the receptor activity in GPCR is significantly enormous. We have trided to compare available information to our observation and then further concentrated our in-vitro studies by mutating selcected amino acids. In human Met134^2.53^, Asn138^2.57^, Asn142^2.61^, Leu232^E2.49^, Tyr315^6.51^, Phe342^7.39^, and Met346^7.43^ have been reported to be crucial for the binding of NKB and NKB antagonists [64]. Ala mutation at their positional homologs in tilapia Met74^2.53^, Asn78^2.57^, Asn82^2.61^, Ile175^E2.49^, Met285^7.39^, and Met289^7.43^ (the molecules are numbered as per Ballesteros-Weinstein [70] numbering, Spreadsheet S5) showed significant reduction in stability of the receptor with exception of Tyr257^6.51^ which showed a significant increase in receptor stability in the case of tiNKF. Simultaneous mutation to human homolog, Ile175^E2.49^Leu and Met285^7.39^Phe in tiTac3Ra showed a significant reduction in Δ Affinity and Δ Stability of complex with native ligands in tilapia. While Cys98^3.25^Ala on tiTac3Ra showed increases in affinity towards both tiNKB and tiNKF, Asn102^3.29^Ala displayed an increased binding affinity to tiNKB and a similar increase was seen by the His191^5.39^ Ala mutation towards tiNKF. Conserved tiTac3Ra Met289^7.43^ homologs in huNK_3_R Met346^7.43^ [64] and huNK_2_R Met297^7.43^ [65] have been reported to be crucial for the binding affinity of NKB, whereas residue Phe^6.44^ (tiTac3Ra Phe251^6.44^) is well conserved in the GPCRs and forms the hydrophobic core [71] with the CWxPY region. Similarly located conserved Phe^6.43^ on SMO (Smoothened GPCR receptor) of the Class F GPCR family has also been reported to be key to maintain the helical geometry of the GPCR TM6 and crucial for cholesterol-mediated receptor activation [72]. A Pro^6.43^Phe mutation was observed to affect the receptor activity significantly [72]. It also plays a role in the conformational rearrangement of the transmembrane domain in class A GPCR as a part of the effect of the PIF amino acid trio (Pro^5.50^, Ile^3.40^, and Phe^6.44^) on simulant/ligand docking to the receptor [66]. Although Phe251^6.44^ and Met289^7.43^ showed no direct interactions in the binding of the native peptides or with the antagonists in tiTac3Ra, mutations in these amino acids were found to significantly affect the stability of the receptor (Δ stability: Phe251 Ala: 12.96 kcal/mol & Met289Ala: 15.11 kcal/mol). We further used Conserf Server [46] to identify conserved residues (Appendix A) and compared them to our residue mutation results, to understand the role of these residues in the binding and receptor activity. To better understand the role of the tiTac3Ra Met289^7.43^ and tiTac3Ra Phe251^6.44^ in receptor stability and function and to verify our in silico predictions, we further performed an *in-vitro* analysis, using a receptor with Ala mutation at these amino acid positions.

Our *in-vitro* luciferase assay showed that the receptor activity was decreased drastically for the mutants, indicating that Phe251^6.44^ and Met289^7.43^ clearly play important roles in simulant-induced receptor activity, and are located in the proposed binding regions, although they do not show any direct interaction with the tiNKB and tiNKF peptides. Due to the lack of any available crystal structure, the studies depicting role of these residues in binding are generally based on effect of their loss of function mutation on downstream receptor signaling, based on our in-silico observation and the position of these residues on the transmembrane domain, we propose that Phe251^6.44^ and Met289^7.43^ are not directly involved in the binding of the ligands; rather, they play pivotal roles in the conformational changes in the transmembrane domain, which further induces post binding receptor activity, in turn, leading to signal transduction and hence, justifies the *in-vitro* results.

The binding of ligand to its receptor plays a dominant role in determining the receptor’s activity, and can efficiently be used to control the downstream receptor-induced signaling pathways. However, this signaling, as well as the affinity between the ligand and the receptor, cannot be co-related unconditionally and either or both are affected by factors that are not directly involved in the binding of the ligands or their stimulants. The human homologs of tiTac3Ra M289^7.43^ huNK_3_R Met346^7.43^ and huNK_2_R Met297^7.43^ [64] are suspected to be one of the key players in binding on huNKB, and have a significant role in the receptor’s activity. Nevertheless, our in silico studies show no direct interaction of these residues with the ligands. tiPhe251^6.44^ homologs are well-established as conserved in the CWxPY region; it also plays a major role in the conformational rearrangement of the transmembrane domain as a part of the PIF amino acid trio (Pro^5.50^, Ile^3.40^, and Phe^6.44^) on simulant/ligand docking to the receptor [65,66]. Thus, indications are that these residues do not directly influence the binding of the ligand, but are crucial for the conformational modifications that occur post-docking, and lead to downstream receptor-induced cAMP signaling.

The effect of NKB/NKF peptides on gonadotropin secretion in fish is already well established. The first indication that NKB is widely expressed in fish species and has a stimulatory reproductive effect was shown in zebrafish, goldfish, and tilapia [20,21,22]. Furthermore, tiNKB analog (Succ-Asp-Ile-Phe-N(Me)Ile-Gly-Leu-Met-NH2), zfNKBa analog (Succ-Asp-Ile-Phe-N(Me)Val-Gly-Leu-Met-NH2), zfNKBb analog (Succ-Asp-Phe-N(Me)Val-Gly-Leu-Met-NH2), and NKF analog (Succ-Asp-Ser-Phe-N(Me)Val-Gly-Leu-Met-NH2), significantly increased FSH and LH release both *in-vivo* and *in-vitro* [30]. Osetanant reduced the receptor activity by ~30%, whereas SB-222200 reduced the receptor activity by half. Similar inhibition was observed in the mammalian Tac3R by SB-222200. SB-222200 inhibited NKB binding to the membranes of CHO cells, stably expressing the Tac3R receptor and antagonized NKB-induced Ca^2+^ mobilization in HEK 293 cells [49]. SB-222200 showed a decrease in the LH levels in plasma on solitary IP while administration with sGnRH showed a drop in LH release. Similar results were found in OVX ewes, when micro-implants containing SB-222200, that were placed in the ARC, disrupted LH secretion [73]. However, plasma FSH did not show the same inhibitory effect as was shown for LH plasma levels (Figure 7B). The FSH levels went up 6 h post treatment in all cases and an increase in FSH was seen as soon as 2 h after antagonist administration. In a similar manner, daily oral dosing of the NK3R antagonist, ESN364, throughout the menstrual cycle in nonhuman primates (*M. fascicularis*) did not alter FSH levels [74]. Interestingly, SB-222200 had no antagonistic effect on the Tac3R of the striped bass [25].

Due to a scarcity of evidence for antagonistic activity of SB-222200 and Osanetant, we further performed *in-vivo* experiments to ascertain the effect of antagonists on gonadal development and spermatogenesis. Spermatogenesis is a highly organized and coordinated process, in which diploid spermatogonia proliferate and differentiate to form mature spermatozoa. The process is conserved in vertebrates and morpho-functionally can be divided into three different phases: the mitotic or spermatogonial phase with the different generations of spermatogonia; the meiotic phase with the primary and secondary spermatocytes; and the spermiogenic phase with the haploid spermatids emerging from meiosis and differentiating into spermatozoa [57]. The steroid hormone 11-ketotestosterone (11-KT) is the major androgen in fish and promotes spermatogenesis [57]. This is in accordance with our current results, where 11-KT levels increase gradually as spermatogenesis proceeds and decreases at spermiation in the control group but was very low at the NKB antagonist-injected group (Figure 8A). Experiments with another cichlid, *Astatotilapia burtoni*, showed increased levels of plasma 11-KT, as well as an increased size of the gonad in socially ascending males while subordinated, non-reproducing males showed the opposite effect [75]. In addition to the 11-KT plasma levels, the NKB antagonist affects sperm content and morphology by reducing significantly the mature form of sperm, spermatid, and spermatozoa in tilapia. Tilapia males have significant amounts of Tac3 receptors in the testis [22]. This can explain the effect of NKB on sperm production and the changes of testis morphology in tilapia. In humans it has already been shown that NK3 receptors are located on sperm cells [76].

Unlike the case in humans, fish express NKF apart from NKB and they both play crucial roles in reproductive physiology of the organism. There is a lack of known structures for NKB/NKF receptors and the function of the underlying mechanism for the binding interaction and mechanism. Generally, receptor activity assays analyze different components of cytosolic signal transduction like cAMP, which can suggest residues involved in the signal transduction and yet cannot point out precisely which residue is responsible for binding. In our study we targeted two such residues, Phe251^6.44^ and Met289^7.43^, which are reported to play an active role in binding interaction. Although we observed their significance in receptor activation and signal transduction; in coherence with our in silico data it seems that these residues are more actively involved in the post binding receptor activation.

## 5. Conclusions

In this study we have predicted the binding models and have correlated them to the *in-vitro* and *in-vivo* results of our experiments. NKB antagonists reduced LH secretion and sperm production *in-vivo*. We have focused our *in-vitro* studies on Phe251^6.44^ and Met289^7.43^ of the tiTac3Ra. However, our studies showed no direct interaction between either of these residues to the ligand. We believe these residues to be in post binding conformational modification of the receptor that leads to signal transduction. Phe^6.44^ is well known to be a part of the PIF trio responsible for post-transductional helical rearrangement necessary for the G-protein signaling. In our studies we have also observed tiNKB to be a more efficient ligand despite tiNKF IFD models expressing higher binding affinity. The better response to tiNKB demands further investigation into the post docking molecular dynamics.

## Figures and Tables

**Figure 1 biology-10-00968-f001:**
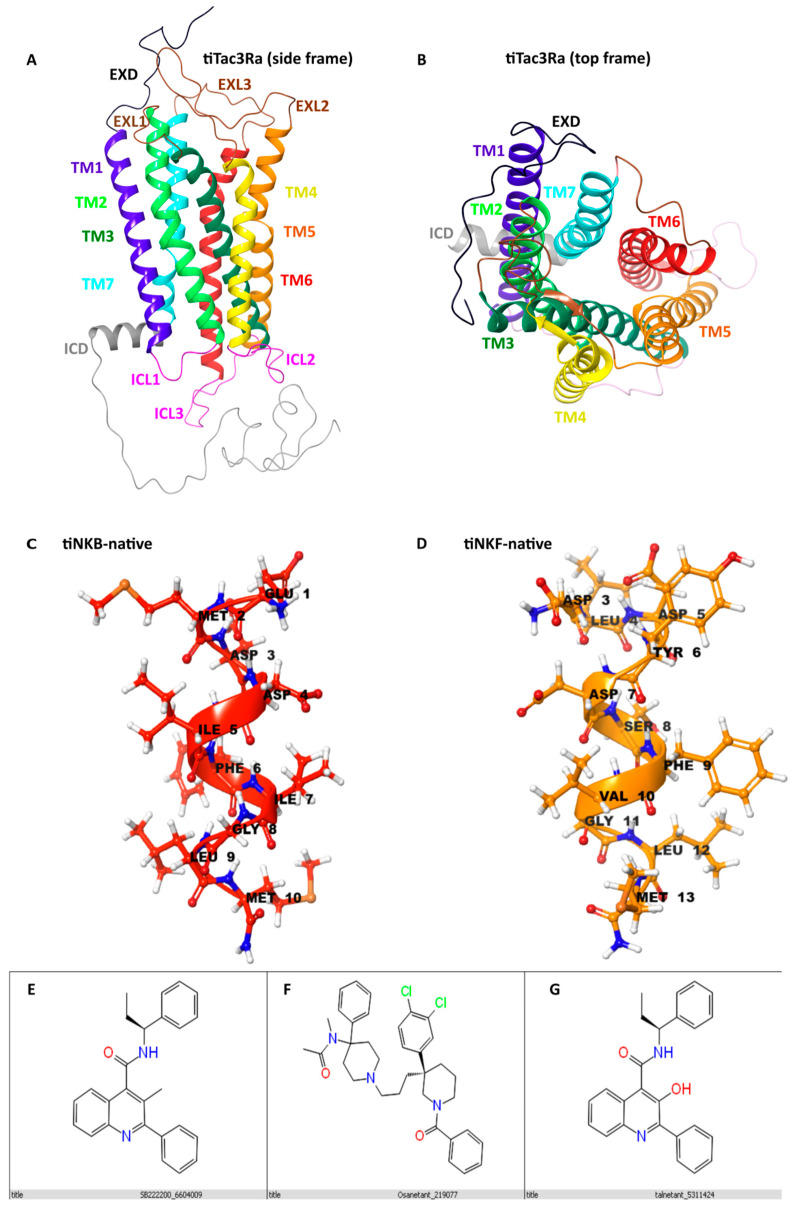
Homology models: Showing predicted homology models for tiTac3Ra, tiNKB & tiNKF and their antagonists (**A**); Side view of tiTac3Ra: extra cellular domain (EXD: Black); Intracellular domain (ICD:Grey); trans membrane domains (TM1: Purple; TM2:Green; TM3:Dark green; TM4:Yellow; TM5: Orange; TM6:Red; TM7:Cyan). Intracellular loops: Pink; Extra cellular loop: Brown. (**B**) Top view tiTac3Ra; (**C**) Native tiNKB; (**D**) Native tiNKF. NKB antagonists: (**E**) SB-222200; (**F**) Osanetant; (**G**) Talnetant.

**Figure 2 biology-10-00968-f002:**
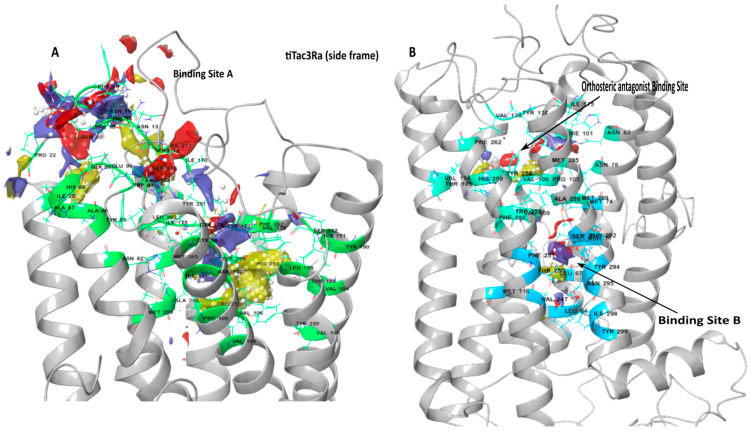
Binding pockets on tiTac3Ra: (**A**). Binding site A; (**B**). Binding Site B & Orthosteric Antagonist Binding Pocket (refer to Table 1: binding site map). The involved amino acids are in green (Site A), blue (Site B) and cyan (Orthosteric binding pocket). The site map shows the hydrophobic region (yellow), the hydrogen-bond donor (blue), and acceptor maps (red).

**Figure 3 biology-10-00968-f003:**
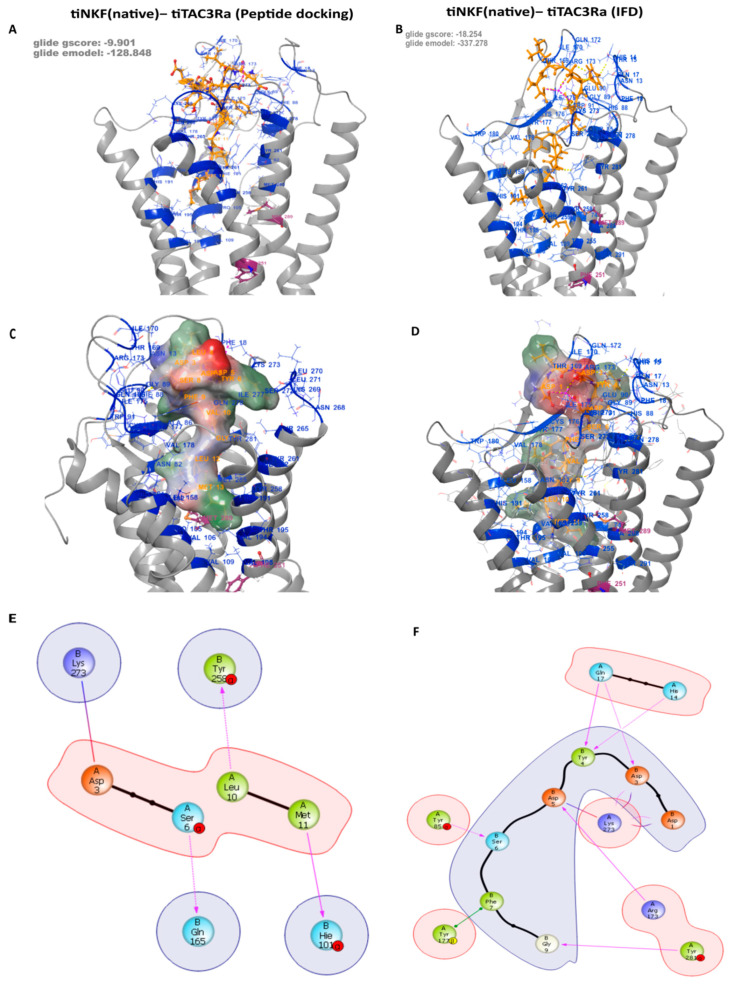
tiNKB/tiNKF docking on tiTac3Ra: Predicted peptide docking of native tiNKB and tiNKF on tiTac3Ra (**A**,**C**,**E**), showing tiNKB showing IFD between tiNKB-tiTacRa (**B**,**D**,**F**). The commonality is a terminal C-terminus with the amino acid sequence Phe—X—Gly—Leu—Met—NH2 docked to the hydrophobic transmembrane pocket.

**Figure 4 biology-10-00968-f004:**
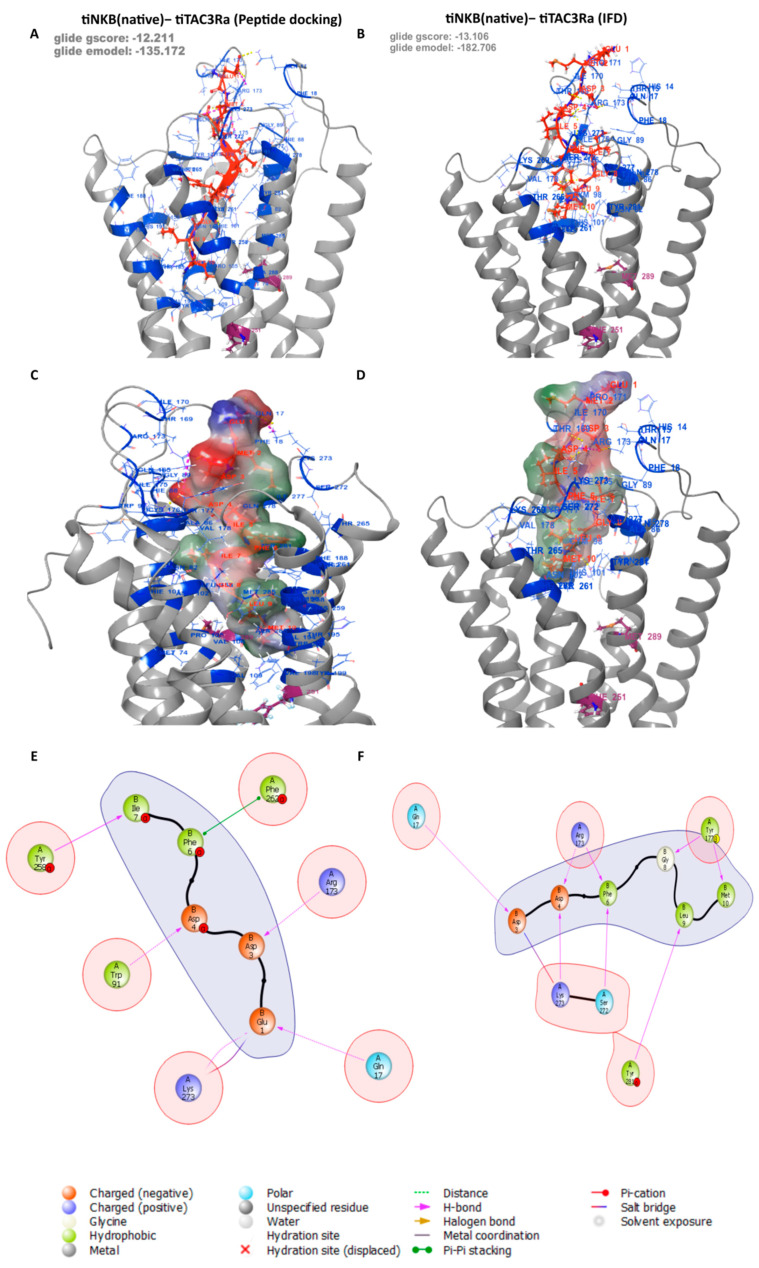
tiNKB docking on tiTac3Ra: Predicted peptide docking of native tiNKB on tiTac3Ra (**A**,**C**,**E**), showing IFD between tiNKB-tiTacRa (**B**,**D**,**F**) The commonality is a terminal C-terminus with the amino acid sequence Phe—X—Gly—Leu—Met—NH_2_ docked to the hydrophobic transmembrane pocket.

**Figure 5 biology-10-00968-f005:**
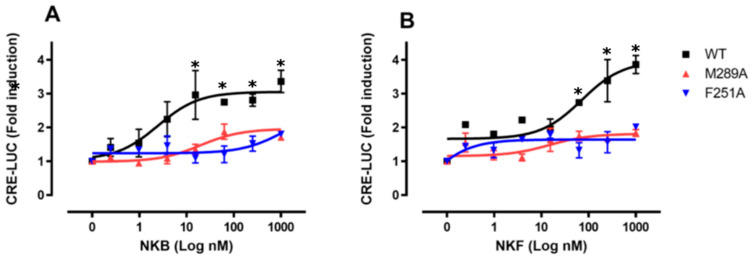
Luciferase assay (Mutation on tiTac3Ra): COS-7 cells were transfected with the native tilapia Tac3Ra and the mutant Tac3Ra, M289A, and F251A, along with reporter plasmid CRE-luc. The cells were treated with various concentrations of tiNKB (**A**) and tiNKF (**B**). The data are expressed as the change in luciferase activity over basal activity, and are from a single experiment, representative of a total of three such experiments. Each point was determined in triplicate and is given as a mean ± SEM. Asterisk (*****) represent significant differences (*p* < 0.05; One way ANOVA followed by Dunnett).

**Figure 6 biology-10-00968-f006:**
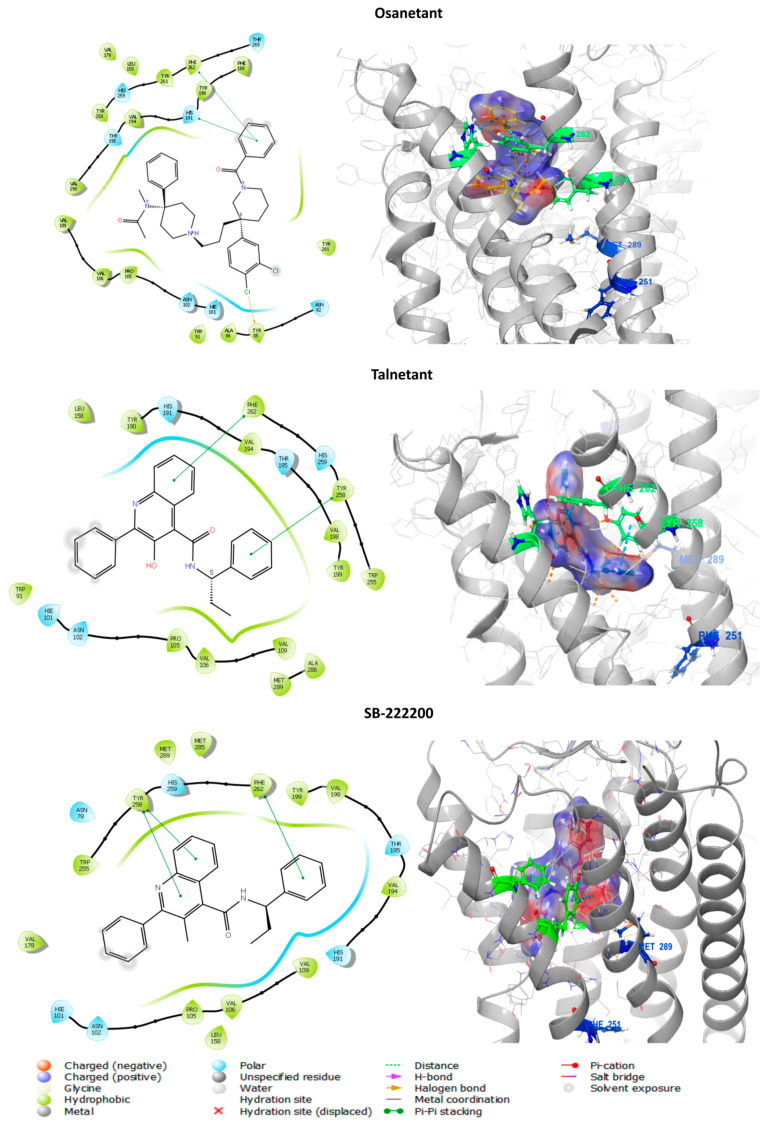
NKB antagonists docking on tiTac3Ra: Predicted docking of NKB antagonists Osanetant, Talnetant, and SB-222200.

**Figure 7 biology-10-00968-f007:**
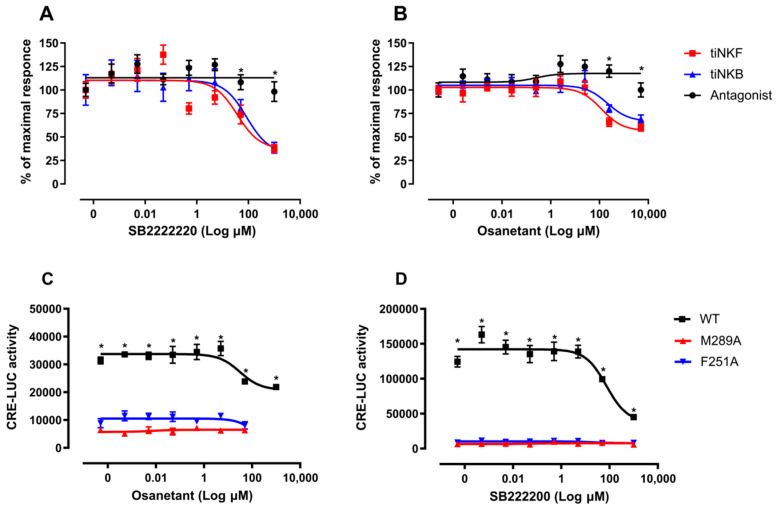
Luciferase assay (antagonists): COS7 cells were transfected with tiTac3Ra together with reporter plasmid CRE-luc. The cells were treated with various concentrations (0.001–1000 μM) of NKB antagonists, SB-22220 (**A**) and Osanetant (**B**) together with constant concentration (10 nM) of tiNKB or tiNKF. The data are expressed as percentage of maximal response and are from a single experiment, representative of a total of triplicates. Each point was determined in triplicate and is given as a mean ± SEM. COS7 cells were transfected with native tiTac3Ra and the mutant tiTac3Ra, M289A, and F251A, along with reporter plasmid CRE-luc. The cells were treated with various concentrations (0.001–1000 μM) of NKB antagonists, SB-222220 (**C**) and Osanetant (**D**) together with constant concentration (10 nM) of tiNKB. The data are expressed as % of maximal response and are from a single experiment, representative of a total of three such experiments. Each point was determined in triplicate and is given as a mean ± SEM. Asterisk (*****) represent significant differences (*p* < 0.05; One way ANOVA followed by Dunnett).

**Figure 8 biology-10-00968-f008:**
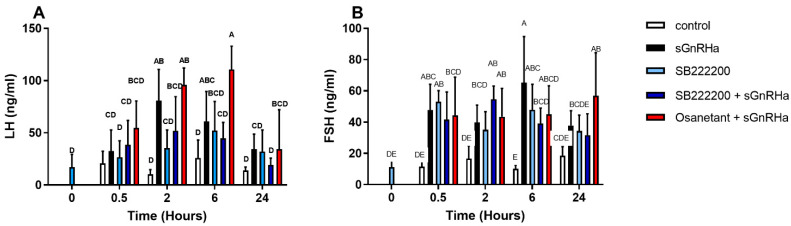
Showing In vivo effect of NKB antagonists, SB-222200, and Osanetant, on LH (**A**) and FSH (**B**) plasma levels. Male tilapia were IP injected with NKB antagonists (100 µg/kg BW) and after 0.5 h, a second injection of sGnRHa ([DAla^6^,Pro^9^-Net]-mammalian GnRH; Bachem) 10 µg/kg BW) was added. Control groups were injected with sGnRHa (10 µg/kg BW) as a positive control, and saline. Plasma LH and FSH values were analyzed by specific ELISA (Mean with 95% CI; *n* = 10 fish per group). Columns marked with different letters are significantly different (two-way ANOVA followed by Tukey HSD).

**Figure 9 biology-10-00968-f009:**
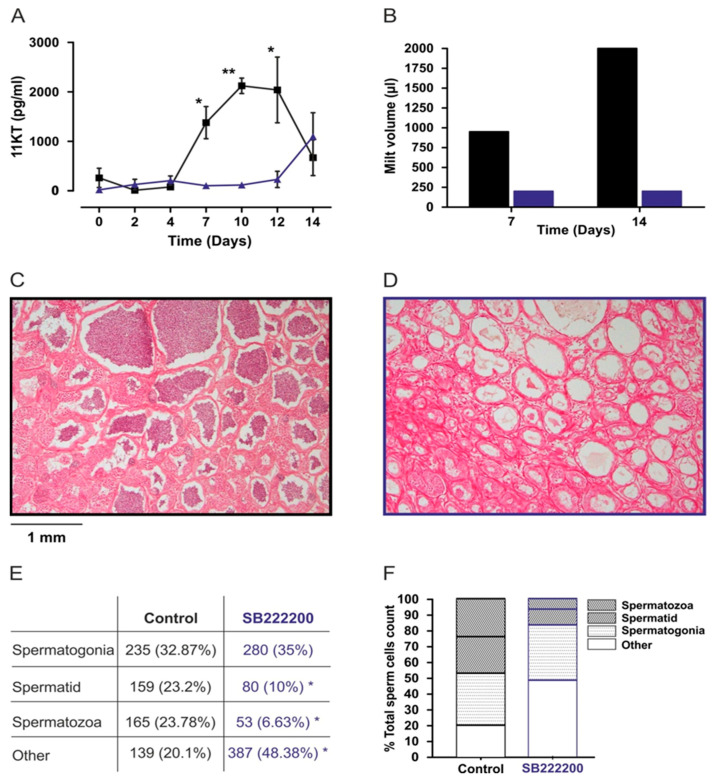
In vivo effect of NKB antagonist, SB-222200, (Blue) compared to control group (Black) on sperm production and volume. Male tilapia were IP injected with SB-222200 (500 µg/kg BW in 25% DMSO), and saline with 25% DMSO as a control every 48 h during a 14-day period. Blood was sampled every 48 h for 11-KT ELISA analysis (Mean with 95% CI; *n* = 6 fish per group (**A**). Milt was sampled at 7th and 14th day of injections and its volume was measured. (**B**). At 14 days, testes were sampled for histology from control fish (**C**) or SB-222200 injected fish (**D**). Different stages along spermatogenesis were estimated (**E**,**F**); * *p* < 0.05, ** *p* < 0.01.

**Table 1 biology-10-00968-t001:** Primers for modification of tiTac3Ra mutant expression vector.

Primer	Position	Sequence (5′ 3′)	Use
tiTac3Ra -F251A-F	732	GATTGTTGTAGTGGTGACGGCCGCCCTCTGTTGGCTGCCAT	F251A mutation
tiTac3Ra- F251A-R	732	ATGGCAGCCAACAGAGGGCGGCCGTCACCACTACAACAATC	F251A mutation
tiTac3Ra- M289A-F	843	CCTGTCGATCATGTGGCTTGCAGCCAGTTCCACCATGTACAACC	M289A mutation
tiTac3Ra- M289A-R	843	GGTTGTACATGGTGGAACTGGCTGCAAGCCACATGATCGACAGG	M289A mutation

**Table 2 biology-10-00968-t002:** List of Amino acids exposed to the Predicted Binding Sites on tiTac3Ra. (refer Figure 2).

Binding Site	Involved Amino Acid on tiTac3Ra
Binding pocket A	Thr12, Asn13, His14, Thr15, Asn16, Gln17, Phe18, Val19, Gln20, Pro22, Ile25, Asn82, Tyr85, Ala86, Ala87, His88, Gly89, Glu90, Trp91, Cys98, His101, Asn102, Pro105, Val106, Val109, Leu158, Ser162, Ile170, Arg173, Ile175, Cys176, Tyr177, Val178, Tyr190, His191, Val194, Thr195, Val198, Tyr199, Trp255, Tyr258, His259, Phe262, Ile277, Gln278, Tyr281, Leu282, Met285, Ala288, Met289.
Binding pocket B	Leu64, Leu67, Asp71, Met74, Met116, Val247, Thr250, Phe251, Ser291, Thr292, Tyr294, Asn295, Ile298, Tyr299.
Orthosteric Antagonist binding pocket	Asn78, Phe82, His101, Pro105, Val106, Val109, Phe110, Ile175, Tyr177, Val178, Val194, Thr195, Trp255,Tyr258, His259, Phe262, Met285, Ala288, Met289.

**Table 5 biology-10-00968-t005:** Dose response in vitro analysis.

	NKB			NKF		
Mutation	WT	F251A	M289A	WT	F251A	M289A
EC50 (nM)	2.48 ± 0.05	X	X	65.22 ± 18.43	X	X
Min effective dose (nM)	0.316	X	X	0.316	X	X
Max effect (Luc fold activation)	3.36 ± 0.46	2.0 ± 0.32	1.71 ± 0.46	3.87 ± 0.38	1.99 ± 0.04	1.84 ± 0.17

EC50 (nM), minimal effective dose (nM) and maximal effect (Luc fold activation) for the native tilapia Tac3Ra and for the Tac3Ra mutants, F251A and M289A. EC50 values and minimal effective doses were calculated when maximal effect (Luc fold activation) was >2.0. (Data from Figure 4).

## Data Availability

Not applicable.

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
