# Peer review of "Characteristics of Neurokinin-3 Receptor and Its Binding Sites by Mutational Analysis"

_biology, 2021, doi:10.3390/biology10100968_

Round 1
Reviewer 1 Report
Please see attachment.

Author Response
Reviewer 1
Simple Summary
- All the acronyms that be given at the first time, have to be give the whole word in detail.
Ans: The reviewer's suggestions have been implemented as necessary.
Summary /Abstract
Suggests improving the abstract so that it is clear to the reader of the purpose of the study, methods, results and conclusions.
Ans: The reviewer's suggestions have been implemented as necessary.
Introduction
- Please describe in more detail the various GnRHs (1,2, and 3) and their role in fish reproduction.
Ans: The reviewer's suggestions have been implemented as necessary. Line 55-63
- I suggest first describing all the hormones and receptors in the axis (HPG). And then to describe what is known in fish and mammals about Neurokinin-3 Receptor. I think it will be easier for the reader to understand.
Ans: The reviewer's suggestions have been implemented as necessary.
- Please add the names of the fish species (In fish NKB was shown to stimulate the release and synthesis of LH and FSH both in-vivo and in-vitro from several species [15, 17, 20, 25, 26]).
Ans: According to the reviewers' suggestions, names of fishes were added to the sentences based on the references.
- Is this sentence is correct in all fish? (However, fish GnRH neurons do not possess kisspeptin receptors). Please clarify the sentence.
Ans: As is supported by the reference provided, it seems to hold true for teleosts such as medaka (Oryzias latipes), goldfish (Carassius auratus), European sea bass (Dicentrarchus labrax), and African cichlid (Astatotilapia burtoni). References have been added to indicate that the case is such in these species.
- Please explain in detail - what is an important reproductive regulator.
Ans: We have shown that in tilapia, NKB is an important neuropeptide that regulates FSH and LH release (Biran et al., 2014, Mizrahi et al., 2019). Unlike mammals where kisspeptin plays a more crucial role.
- Please define more detail the specifics aims of the study.
Ans: The reviewer's suggestions have been implemented as necessary. Line 125
Materials and Methods
- Fish maintenance and growth – Please add more accurate details about the fish: age, weight, etc ... Please add reforms that support the methods (Transient transfection, cell procedures ...... each performed in triplicate. )
Ans : In this section we are mentioning general fish maintenance and growth for all of our in vivo experiments. Specific details about the fish are presented under Sections 2.8 and 2.9.
- 9 In-vivo effect of NKB antagonist on sperm production in tilapia – add reference.
Ans: Reference No 51(Heyrati, F. P.; Amiri, B. M.; Dorafshan, S., Effect of GnRHa injection on milt volume in recently stripped rainbow trout Oncorhynchus mykiss. Aquaculture Research 2010, 41, (10), e487-e492.) was added
- 11 11-ketotestosterone (11KT) analysis – add reference Add statistical analysis (e.g. Fig.5).
Ans: the ELISA for 11KT is according to reference #50 (Aizen et al., 2007) – this was corrected in the text.
Results
- I suggest adding in the titles of the illustrations of the models the references of the method by which the model was constructed.
Ans: Titles and references were added as suggested.
- 5, 7, 8 and 9 - add statistical differences among the means and the Curves.
Ans: We added statistical analyses for figs 5 and 7. Figs 8 and 9 already contains the respective statistical analysis.
Discussion
- The first paragraph states that there is not much information and no references and citations of studies have been presented ?
#Methods Mol Biol .0202;145:050-16 . doi: 10.1007/978-1-60761-762-4_8. Recent progress in the structure determination of GPCRs, a membrane protein family with high potential as pharmaceutical targets Vadim Cherezov 1, Enrique Abola, Raymond Stevens
#COMMENTARY The quest for high-resolution G protein-coupled receptor–G protein structures. Reinhard Grisshammer
#Structural and functional characterization of G protein–coupled receptors with deep mutational scanning
Ans: The paragraph talks specifically about unavailability of structures of NKB receptors and their binding mechanisms as there are none available to our knowledge, and not GPCRs in general. Sentence structure was amended to avoid misunderstanding. References were added as necessary.
- The discussion has a lot of information but lacks the simple general structure of a discussion that allows the reader to understand the following points. The main contribution of the article compared to other works. Criticism of measurement methods and suggestions for improvement. Suggestions for further research in the same field.
Ans: The reviewer's suggestions have been implemented as necessary.
Reviewer 2 Report
Line 51, citation [2]: This citation is following the sentence “Hence, the teleosts anterior pituitary is innervated by neurons synthesizing a number of neuropeptides and neurotransmitters involved in the regulation of the gonadotropins (GTHs), luteinizing hormone (LH), and follicle-stimulating hormone (FSH) release [2].” But the paper is on the control of pars distalis (posterior pituitary or neurohypophysis) by the hypothalamus. One of two things, or the citation is wrong or the authors meant posterior pituitary. Please, clarify it.
Line 166: Spreadsheets S3 & S4 should be numbered S1 & S2, given they are the first to appear in the text, before the named S1 & S2. Moreover, they have no content, except the headings. Check well that.
Line 510: In the Figure 9 legend, the description for graph (B) “Milt was sampled at 7 and 14 days (B).” is not enough. You need to describe what is in Figure 9B that is not so obvious: Milt was sampled at 7 and 14 days, and its volume was measured. Black and blue bars meaning from the very beginning. It is not stated in any of the subgraphs legends.
Lines 693, whole 5 Conclusion: The conclusion should be a few sentences straight to the main findings of the research, without discussion or comparisons. There is a lot of comparisons and discussion in this part that should be avoided.
Line 884: in Reference 60 appears a [19] which meaning is unknown. Delete it.

Author Response
Reviewer 2
- Line 51, citation [2]: This citation is following the sentence “Hence, the teleosts anterior pituitary is innervated by neurons synthesizing a number of neuropeptides and neurotransmitters involved in the regulation of the gonadotropins (GTHs), luteinizing hormone (LH), and follicle-stimulating hormone (FSH) release [2].” But the paper is on the control of pars distalis (posterior pituitary or neurohypophysis) by the hypothalamus. One of two things, or the citation is wrong or the authors meant posterior pituitary. Please, clarify it.
Ans: Pars distalis is the distal part of anterior pituitary and forms majority of the same; not to be confused with Pars nervosa which forms major part of posterior . Sentence was reframed to avoid confusion.
- Line 166: Spreadsheets S3 & S4 should be numbered S1 & S2, given they are the first to appear in the text, before the named S1 & S2. Moreover, they have no content, except the headings. Check well that.
Ans: We thank the reviewer for pointing this out. Changes have been made as necessary. The files have been changed appropriately.
- Line 510: In the Figure 9 legend, the description for graph (B) “Milt was sampled at 7 and 14 days (B).” is not enough. You need to describe what is in Figure 9B that is not so obvious: Milt was sampled at 7 and 14 days, and its volume was measured. Black and blue bars meaning from the very beginning. It is not stated in any of the subgraphs legends.
Ans: The reviewer's suggestions have been implemented as necessary.
- Lines 693, whole 5 Conclusion: The conclusion should be a few sentences straight to the main findings of the research, without discussion or comparisons. There is a lot of comparisons and discussion in this part that should be avoided.
Ans: The reviewer's suggestions have been implemented as necessary.
- Line 884: in Reference 60 appears a [19] which meaning is unknown. Delete it.
Ans: We thank the reviewer for pointing this out. Corrections have been made to the text as suggested.